# Ocular Complications of Giant Cell Arteritis: An Acute Therapeutic Emergency

**DOI:** 10.3390/jcm11071997

**Published:** 2022-04-02

**Authors:** Emmanuel Héron, Neila Sedira, Ouassila Dahia, Céline Jamart

**Affiliations:** Department of Internal Medicine, Centre Hospitalier National d’Ophtalmologie des Quinze-Vingts, 28 Rue de Charenton, 75012 Paris, France; nsedira@15-20.fr (N.S.); odahia@15-20.fr (O.D.); cjamart@15-20.fr (C.J.)

**Keywords:** giant cell arteritis, ophthalmologic manifestations, diagnosis, treatment

## Abstract

The risk of blindness, due to acute ischemic ocular events, is the most feared complication of giant cell arteritis (GCA) since the middle of the 20th century. A decrease of its rate has occurred after the advent of corticoid therapy for this vasculitis, but it seems to have stabilized since then. Early diagnosis and treatment of GCA is key to reducing its ocular morbidity. However, it is not uncommon for ophthalmological manifestations to inaugurate the disease, and the biological inflammatory reaction may be mild, making its diagnosis more challenging. In recent years, vascular imaging has opened up new possibilities for the rapid diagnosis of GCA, and ultrasound has taken a central place in fast-track diagnostic processes. Corticosteroid therapy remains the cornerstone of treatment and must begin immediately in patients with visual symptoms and suspicion of GCA. In that situation, the administration route of corticotherapy, intravenous or oral, is less important than its speed of delivery, any hour of delay worsening the prognosis.

## 1. Introduction

In the 1950s, ocular complications of GCA, called at that time temporal arteritis (TA), emerged as its more common serious complication, with a risk of partial or total blindness in about half of the patients. Most knowledge about the association between giant cell arteritis (GCA) and its ocular complications, their mechanism, frequency, diagnosis, prognosis, relationship with other manifestations of the disease, and the potential benefit of corticosteroid therapy has been described during this period of time. We briefly review the pathological, clinical, and therapeutics aspects of this condition which became 70 years ago the most acute medical emergency in ophthalmology and remains so. This review makes room for historical data, as they usefully complement the most recent ones.

## 2. From a Benign Disease of the Temporal Arteries to an Acute Ophthalmological Emergency: A Brief Historical Overview

In 1932, Horton, Magath, and Brown described the first two cases of temporal arteritis (TA), as a new distinct clinical entity [1], which was initially thought to be localized, self-limited, and benign. By the early 1940s, TA was recognized as a generalized vascular disease, as histological lesions identical to those seen in the temporal arteries were also found in autopsy cases in the retinal, aorta, carotid, subclavian, pulmonary, and many other visceral and peripheral arteries [2,3]. The description of the systemic symptoms of the disease, i.e., fever, anorexia, malaise, weakness, and loss of weight, was added to the previously described cranial symptoms: tender, swollen, nodular temporal arteries, scalp hyperesthesia, headache, jaw claudication, and ocular manifestations [4]. It was recognized that the disease may be a rare cause of death, particularly from stroke [4]. Since this vasculitis was not limited to temporal arteries, other names than TA were proposed, among which that of giant cell arteritis (GCA) in 1941 for the first time [5]. The first case of TA-associated unilateral blindness was published in 1938 by Jennings [6], while Horton and his Mayo Clinic colleagues reported cases of TA associated to vision loss only in 1943 [7]. During this decade, the risk of sudden and irreversible blindness has been recognized as the most common and feared threat of the disease, and its main causes have been identified as ischemic optic neuropathy and central retinal artery occlusion [8,9]. In 1948, Chavany and Taptas were the first to report dramatic improvement in a TA patient on corticosteroid therapy [10]. Since October 1949, most GCA patients seen at the Mayo Clinic were started on cortisone or adrenal corticosteroids [11]. During the 1950s, corticosteroids emerged as the reference treatment for TA [12,13,14], with the capacity to decrease its ocular morbidity, and this remains today.

## 3. Ocular Manifestations of GCA

### 3.1. Histopathological Data

The ocular manifestations of GCA are ischemic in origin and as such are closely related to the location and severity of the cephalic vascular lesions associated with the disease [15]. The primary mechanism of ocular ischemia in GCA is the narrowing of the arterial lumen by the arteritic process, with a risk of occlusion inversely proportional to the arterial caliber. Thrombosis is neither a constant nor a characteristic feature of the disease. For example, it has been found in only 1/3 of affected biopsied temporal arteries [11]. However, in certain cases, it could precipitate the occurrence of visual loss. Moreover, thromboembolism appears to be the first cause of GCA-related strokes [15].

It is of great interest for the clinician to detail the histopathological data on this subject. The arterial lesions of the head and neck have been precisely described in 12 autopsy cases of GCA patients who died during the active phase of the disease between 1940 and 1971 [15]. The highest incidence (75–100%) and most severe lesions were found in the temporal, vertebral, ophthalmic, and posterior ciliary arteries. They were less frequent (35–60%) and generally less severe in the internal carotid (petrous and cavernous segment), other external carotid branches (facial, occipital), and central retinal arteries. Of note, the intracranial arteries were never involved. In this series, close clinicopathological correlation was found between clinical events before death, i.e., sudden unilateral blindness in seven cases, sudden occipital blindness in two, signs of brain stem ischemia in five, and of hemispheric cerebral ischemia in two, and autopsy findings showing ischemic damage in the optic nerves in 7 (associated with retinal ischemia in 2), occipital regions in 2, brain stem and cerebellum in 6, and cerebral hemispheres in 4 patients. Brain ischemia was of embolic origin, according to the pathological findings. Remarkably, the histological involvement of the vertebral arteries, although severe in the neck, always ceased abruptly 5 mm after the passage of the dura. Similarly, the internal carotid artery was never involved beyond the dura. The description of the histological lesions of the central retinal artery is also very interesting. This artery was markedly involved in its course outside the optic nerve, much less in its short segment within the dural sheath of the nerve, and absent when it runs in the optic nerve. Branches of the artery within the eyeball were always free of lesions. Thus, with regard to the head and neck, arterial changes in GCA patients were lacking in the brain, the optic nerve and the eyeball. This was in marked contrast to the constant and severe lesions of orbital arteries outside the optic nerve sheath, particularly the ophthalmic and posterior ciliary arteries, and of cervico-cranial arteries outside the skull, mainly the vertebral and temporal ones. In explanation of this fact, there was an obvious correlation between the presence and severity of the arterial lesions and the amount of elastic tissue in the media and adventitia in the artery walls. Indeed, the intracranial arteries and the orbital arteries located inside the optic nerve sheath and the eyeball have little or no elastic tissue [15]. These histopathological data described in 1972 were again observed in more recently reported autopsy cases [16,17].

### 3.2. Causes of Permanent Visual Loss

The type and relative frequency of the ophthalmic manifestations of GCA described in the literature [18,19,20,21,22,23] correspond well to these historical pathological data. The causes of visual loss are listed in Table 1, according to the anatomical location of the lesions. In more than 80% of cases, permanent visual loss is due to acute ischemic optic neuropathy (ION), mostly anterior (AION) by occlusion of short posterior ciliary arteries and sometimes posterior (PION) by occlusion of nutrient arteries of the nerve trunk, arising from the ophthalmic artery. Since the posterior ciliary arteries also give out the cilioretinal artery when there is one (25% of patients) and supply the choroid, their occlusion may induce cilioretinal artery occlusion and choroidal ischemic lesions [18]. The latter may appear on fluorescein angiography performed soon after the acute event, as delayed filling or non-perfused areas of the choroid. The second most frequent cause is central retinal artery occlusion, in around 10% of cases. The third rare cause is occipital blindness secondary to a vertebro-basilar stroke.

Other rare ophthalmological presentations include anterior and/or whole ocular ischemia, with a risk of neovascular glaucoma, involving the anterior ciliary artery territory, and orbital inflammatory syndrome, involving orbital branches of the ophthalmic artery and granulomatous intra-orbital inflammatory reaction. Since intra-ocular branches of the central retinal artery are not histologically involved [15], there is no reason to find occlusion of such branches in isolation in GCA.

### 3.3. Transient Visual Manifestations

Transient ophthalmic manifestations are quite frequent in GCA, including amaurosis fugax and diplopia, both of which should alert to a risk of impending permanent visual loss. Amaurosis fugax may reflect optic nerve or retinal ischemia, probably most often from low flow in the narrowed upstream arteries, rather than from microemboli [18]. Diplopia has been a subjective symptom of short duration in most reported cases; otherwise, it always cleared under treatment. Basically, diplopia could be due to the arteritic involvement of nutrient arterial branches arising from the ophthalmic artery and supplying to the oculomotor nerves or muscles [11,18]. Wagener and Hollenhorst examined three GCA patients with persistent oculomotor pareses and observed a considerable daily variation in the muscles involved and in their degree of impairment, suggesting a kind of “ocular claudication” [11]. Such observation gives support to a myogenic rather than neurogenic mechanism of diplopia in GCA patients, but both could probably happen. In their series of 122 GCA patients collected at the Mayo Clinic until 1956, these authors observed amaurosis fugax in 20 cases (16%), followed by loss of vision in 16 (80%), and diplopia in 12 cases (10%), preceding visual loss in “a number of instances” [11]. A bit later, Whitefiel et al. found 12 cases of diplopia among 72 GCA patients (17%), and 7 of them (58%) developed visual loss, while 8 patients had amaurosis fugax, preceding visual loss in three [24]. In the corticosteroid era, in a study of 170 GCA patients, 85 with ocular manifestations, amaurosis fugax occurred in 26 of them (15%) and diplopia in 5 (3%), followed in both cases by permanent visual loss in two-thirds of the patients [18]. Thus, in patients over the age of 50, the primary emergency in case of amaurosis fugax or diplopia is to rule out GCA or diagnose and treat it immediately to save sight.

### 3.4. Time Element of the Development of Ocular Manifestations

In their historical series from the Mayo Clinic, Wagener and Hollenhorst reported that, among 54 GCA patients with visual loss, ophthalmic symptoms heralded the disease in 2 of them, and in the others, they occurred at intervals varying from less than one day to eleven months after the first extra-ophthalmological symptoms, more rapidly in the presence of headache or inflamed temporal arteries [11]. Hollenhorst et al. later reported in 104 patients that this time interval was less than one week in 4%, one week to one month in 26%, one to three months in 60%, and more than three months in 10% of the cases [25]. In a French cohort of 302 biopsy-proven GCA patients including 44 with visual loss, the mean delay between ocular and systemic signs was 43 days (2–230), and in only 17% of cases, it was less than 2 weeks [26]. Thus, the mean interval between general and ocular symptoms is around 6 weeks. This is in agreement with the mean delay of 7.7 weeks from symptom onset to the diagnosis of cranial GCA found in a recent meta-analysis [27].

Birkhead et al., among 108 patients with visual loss, were unable to establish any correlation between a particular systemic sign and the occurrence of ocular damage [14]. During the corticosteroid era, the factors most often associated with an increased risk of permanent visual loss were transient visual episodes and lower levels of biological inflammatory parameters, while constitutional symptoms decreased ocular risk [21,22,28,29]. Other factors such as jaw claudication, abnormal temporal artery, male sex, older age, thrombocytosis, and anemia have been variously associated with the ocular risk [19,21,22,28,29,30,31]. This has little practical significance anyway, since the only real issue is not to miss the diagnosis of GCA and not to try to predict which GCA patient has a higher risk of ocular complications.

Furthermore, it has been recognized very early that the ophthalmological manifestations of GCA could be inaugural in the disease. This was observed in 2 out of 54 patients (4%) in the series of Wagener [11], 5 out of 50 (10%) in that of Parsons-Smith, who also specified that the other signs of GCA developed 3 weeks to 6 months after the ocular signs [32], and 5 out of 32 (16%) in that of Simmons, who called clinically silent, biopsy-proven GCA, “occult temporal arteritis” [33]. More recently, Aiello et al. reported occult TA in 9 out of 34 patients with ocular damage (26%) [19], Liu et al. in 8 out of 41 (20%) [20], Hayreh et al. in 18 out of 85 (21%) [34], and Chen et al. in 5 out of 20 patients (20%) in a population-based cohort [35].

This knowledge is of great importance to ophthalmologists, who first see these patients. It must be emphasized that too much weight should not be given to the presence of systemic signs of the disease, nor to the magnitude of biological inflammatory reaction, in the diagnosis of ocular forms of GCA. Both may be unremarkable, or even lacking, which should encourage further investigation (imaging, biopsy) rather than stopping it.

## 4. Frequency of Permanent Visual Loss

In 1953, Whitfield et al. indicated that more than 150 GCA cases had been published since the description of the disease and that the frequency of visual impairment in five series published between 1946 and 1951 ranged from 33% to 55% [13]. The rates of permanent visual loss in several large retrospective studies, covering different periods of time before and after the routine use of corticosteroid therapy for GCA [9,14,19,21,22,29,30,32,35,36], are shown in Table 2. The largest series in the pre-corticosteroid era reported a rate of permanent visual loss of 40 to 48% of cases, involving both eyes in more than half of them and leading to complete blindness in a majority of the affected eyes [9,14,19].

The study by Birkhead et al. was the first to show a lower rate of bilateral blindness at patient discharge in 55 patients who received steroid treatment at the time of ocular event, compared with 53 patients not treated with cortisone in the same institution [14]. The authors specified that no patient in the treated group developed bilateral blindness during treatment, while bilateral blindness increased from 6 on admission to 17 at the time of discharge in the historical untreated group, and they recommended the generalization of corticosteroid therapy for all GCA patients as long as the active phase endures.

During the corticosteroid era, the rate of definitive visual loss fell approximately from 45% to 15% (Table 2). The lowest incidence (8%), i.e., 20 out of 245 patients who had a new diagnosis of GCA between 1950 and 2009, was found in a retrospective population-based cohort in Olmsted County, Minnesota [35]. In that population, the rate of ischemic optic neuropathy also declined over time, from 15% in the 1950–1979 cohort to 6% in the 1980–2004 cohort (*p* = 0.03%). Although the small number of events (N = 17, over 54 years) weakens this observation, it could reflect the generalization of corticosteroid therapy which was adopted locally by the mid 1980s, according to the authors. However, the rate of definitive visual loss is somewhat higher in most studies published during the 1990s, consistently between 10% and 19%. In a French Burgundian cohort of 405 GCA patients seen over 37 years, this rate remained stable at 14% before and after 1995 [26].

Moreover, bilateral ocular involvement at the initial visit remains frequent, ranging from 24 to 56% in five series [18,19,20]. The chronology of the attack in both eyes could be analyzed in Hayreh’s series, in which 31% of patients (27/85) had bilateral visual loss at presentation [18]. Among them, 17% described it as simultaneous, and for 46%, the interval was from one day to one week. Similar time elements had been given in historical Mayo Clinic series [11,14]. Thus, in GCA patients, both eyes can be affected extremely quickly, within hours or days, in two-thirds of cases.

## 5. Diagnosis of GCA in the Presence of Ocular Ischemic Manifestations

The hypothesis of GCA should always come first when facing sudden ocular ischemic manifestations in patients aged over 50, remembering nonetheless that this vasculitis is exceptional before 55 years of age. Relevant symptoms and signs of GCA and the level of biological inflammatory markers should then be rapidly documented. This may be sometimes enough to get a strong suspicion of GCA. However, as stated above, systemic symptoms of the disease are lacking in around 20% of GCA patients presenting with visual symptoms, and biological inflammation can be mild, or even absent. Thus, the suspicion of GCA cannot be ruled out on these parameters alone. On ophthalmological examination, a number of clues can help in the diagnosis. They have been well detailed in Hayreh’s series of 85 patients: (1) chalky white optic disc edema (69% of cases of AION in this series); (2) cilio-retinal artery occlusion in association to AION or CRAO; (3) cotton wool spots in 1/3 of the eyes with visual loss, likely due to retinal microembolisation from larger regional damaged arteries; and (4) evidence of posterior ciliary artery occlusion and/or choroidal ischemia at early fluorescein angiography in AION, and CRAO as well [18]. These ophthalmological signs can help in particular to differentiate arteritic AION from non-arteritic AION.

A reliable new diagnostic tool for GCA is large vessel imaging, acknowledged in recent expert recommendations [37,38,39]. Ultrasonography (US) and MRI of the superficial temporal artery showed the best performances in cranial GCA, with estimated pooled sensitivities in the main studies of respectively 77% and 73%, and pooled specificities of 96% and 88% [40]. US of temporal arteries has been recommended as first line imaging in patients with suspected predominantly cranial GCA and is the pivotal examination in fast-track diagnostic centers that developed recently [41]. In two retrospective studies comparing a fast-track approach including US of temporal and axillary arteries to conventional clinical practice, the accelerated procedure has been associated with a significantly lower rate of visual loss [42,43]. Vascular imaging can certainly be a confirmatory examination for the diagnosis of GCA in trained medical teams, but it has weaknesses such as its reliance on the quality of imaging devices and its operator-dependency. Our 3-year experience with high-resolution 3T MRI of the cranial arteries for this diagnosis in a specialized ophthalmological center has been disappointing, as both sensitivity and specificity were around 60% (unpublished).

Detection of extracranial arteritis, using US, CT, MRI, and/or positron emission tomography (PET) depending on the clinical setting and local facilities, can also confirm a diagnosis of GCA in patients with visual symptoms and is recommended for general disease assessment at the time of diagnosis [37,38,39]. In an ophthalmological setting, imaging should never delay urgent high-dose corticosteroid therapy and should be performed preferably before 3 days of treatment for US and PET [44,45] and 5 days for MRI [46].

In our opinion, histological confirmation of GCA should be obtained whenever possible [37]. The sensitivity of temporal artery biopsy (TAB) for the diagnosis of GCA ranges from 49% to 85% [37]. A meta-analysis gave a pooled sensitivity for TAB of 77.3% [47], which is consistent with our 22-year experience with TAB in a specialized ophthalmology center (unpublished). The sensitivity of TAB appears to be little impaired within 2 weeks of corticosteroid treatment [48,49], and in one study, positive biopsies were still positive after 6 months of treatment in 75% of cases [50].

Thus, in elderly patients presenting with sudden visual symptoms, the diagnosis of GCA will be based on evidence and detail of ischemic ocular lesions on ophthalmological examination, associated with one or more of the following: other relevant clinical manifestations, level of biological inflammatory parameters, vascular imaging results showing arteritis anywhere, and/or histological proof of GCA.

## 6. Treatment of GCA Patients with Ocular Manifestations

The prevention of ocular complications by corticotherapy is effective, since ischemic ocular attacks occur before treatment in more than 95% of the cases [19,22,26]. After a loss of vision, the chief value of corticosteroid treatment is to safeguard the vision that remains. However, ophthalmological status on corticosteroid therapy may sometimes continue to deteriorate or, conversely, rarely improve. The speed of initiation of treatment is a key element in this outcome.

The following personal (unpublished) clinical case illustrates both the worse and the best possible visual outcome in a single patient who received corticosteroid treatment for bilateral arteritic AION. A 76-year-old diabetic man was seen at our emergency ophthalmological unit one evening in January 2014, complaining of sudden visual loss in the right eye 3 days before. He had a six-week history of scalp hyperesthesia, headache, tender temporal arteries, jaw claudication, anorexia, and polyalgia and had had several episodes of amaurosis fugax in his right eye the week before permanent visual loss. The right fundus showed chalky white optic disc edema (Figure 1A). The left fundus also showed slight papilledema of the lower part of the disc. Visual acuity (VA) was 20/200 in the right eye and 20/40 in the left one. As the diagnosis of GCA was obvious; he received immediately a 500 mg intravenous pulse of methylprednisolone, continued the two next days with 250 mg IV methylprednisolone every 12 h together with oral prednisone 60 mg every morning, and then oral prednisone alone 80 mg/day, and aspirin 75 mg/day. Laboratory results received after the start of treatment showed an erythrocyte sedimentation rate of 97 mm in one hour and a C-reactive protein level of 174 mg/L. Right temporal artery biopsy showed typical GCA. Fluorescein and indocyanin green retinal angiographies performed the day after admission showed, on the right side, marked hypoperfusion of the optic disc and choroid, and cilioretinal artery occlusion (Figure 1B,C), and on the left side, milder hypoperfusion of the optic disc, the choroid, and an unusually long cilioretinal artery (Figure 1D).

Two days after admission, VA had decreased to no light perception in the right eye and remained stable in the left. On that day, Goldman perimetry showed a constricted left visual field (Figure 2A). Ten days after admission, the left VA had increased to 20/25, in association with improved visual field (Figure 2B). At last contact with the patient, in November 2016, he had been off corticosteroids for 2 months. He said he felt very well and had no light perception in his right eye but normal vision in his left eye. Thus, the different therapeutic delay between both eyes clearly led to a radically different result, further deterioration to complete blindness for the first affected eye, and significant improvement in the most recently affected one.

A better ophthalmological outcome with earlier steroid treatment has been underlined since the first historical reports [26]. In an observational cohort, 84 GCA patients with visual loss in 114 eyes had been treated with high-dose corticosteroids, either intravenously (N = 41) or orally (N = 43) [51]. VA improved in 13%, but both VA and visual field improved only in 4%, with no difference with regard to the route of steroid administration. The data suggested a better chance of improvement with earlier treatment (*p* = 0.065). In a prospective cohort study of 34 patients with biopsy-proven GCA and visual loss in 40 eyes, all patients received pulse methylprednisolone IV 1 g/d for 3 days followed by oral prednisone 60 to 80 mg/d [23]. VA improved in 15% of eyes, but only 5% had a corresponding improved visual field, while 27% of initially affected eyes showed visual degradation under IV steroids. The mean number of days to treatment was 1.2 days for those who improved versus 2.6 days for those who deteriorated [23]. In another report of 10 patients treated intravenously and 19 treated orally, improved vision was observed if the treatment begun within one day from the onset of visual symptoms, independently of the route of administration [52].

The risk of further visual degradation despite high-dose corticosteroid is also well-known, mainly during the first week of treatment [19,20,22,53]. Aiello et al. [19] reported bilateralization of blindness in 3/34 patients (9%); Liu et al. [20] reported a visual deterioration in 7/39 patients (18%), corresponding to bilateralization in 3 cases and degradation of the initially affected eye in 5; Hayreh et al. [53] reported a deterioration in 9/91 patients (10%), including bilateralization in 1 case and homolateral deterioration in 8, which occurred most often within 5 days of treatment. In the Danesh–Meyer series, 27% of 40 affected eyes deteriorated by 2 or more lines within the first 6 days after the initiation of high-dose IV steroids, and 3 previously uninvolved eyes became affected within 1 day of starting treatment [23].

High-dose corticosteroid therapy is the mainstay of treatment in patients suspected to have ocular symptoms of GCA and must begin immediately. It is agreed that it can be administered intravenously at a dose of 500 to 1000 mg (or 15 mg/kg) of methylprednisolone per day for 1 to 3 days, or orally at 50 to 100 mg (or 1 mg/kg) of prednisone per day [37,39,54]. Intravenous administration may be preferred if it can begin without delay, which will be the case mostly for patients seen at medical institutions. However, the speed of treatment delivery is much more important than its route of administration: the shorter, the better. Routine use of antithrombotic agents is not recommended in GCA patients [37,39,55], although antiaggregating doses of aspirin may be considered during the first month in those with ocular complications, due to a rise in their risk of stroke at that time [30,56]. Tocilizumab (TCZ), a biologic therapy targeting interleukin-6, has shown efficacy as a corticosteroid-sparing agent in GCA and received marketing approval for its treatment. Data on TCZ and ocular manifestations of GCA are scarce. The rapid steroid tapering commonly associated with its use could increase ocular risk in GCA patients treated with TCZ. In the GIACTA study [57], 1 of 142 patients (0.7%) had AAION at the 24th week of treatment, under TCZ 162 mg every other week and prednisone 2 mg/d. In a recent review of 186 GCA patients treated with TCZ and rapid steroid tapering, two patients (1.1%) developed vision loss after TCZ initiation, one had a previous ocular attack and was still under 1 mg/kg/d of prednisone, while the other had received 3 pulses of 500 mg methylprednisolone followed by TCZ monotherapy [58]. The ocular risk for GCA one year or more after the start of corticoid therapy being approximately 1% [19,22,26], close to these preliminary observations under TCZ, no firm conclusion can be drawn to date on the ocular risk for GCA patients treated by TCZ compared to standard corticotherapy. The value of urgently administered TCZ in acute ischemic complications of GCA, in combination with standard steroid therapy, is unknown. A randomized trial (NCT 04239196) is underway to assess its efficacy in combination with high-dose IV methylprednisolone in the emergency treatment of arteritic AION. To date, however, no treatment other than corticosteroids has shown benefit in the treatment of ocular complications of GCA.

## 7. Conclusions

Ischemic eye injury due to GCA is a devastating condition that can result in bilateral blindness within hours. If visual symptoms occur in a patient over 50 years of age, and there is any suspicion of GCA, saving sight is a race against time that allows no delay. High doses of corticosteroids should be administered to the patient immediately, as confirmatory examinations, whether histological or imaging, do not lose sensitivity during the first days of treatment. For ambulatory patients, the intravenous route is likely to cause a delay, and oral therapy should be started in the physician’s office or immediately after his visit, along with urgent referral to a specialized center.

## Figures and Tables

**Figure 1 jcm-11-01997-f001:**
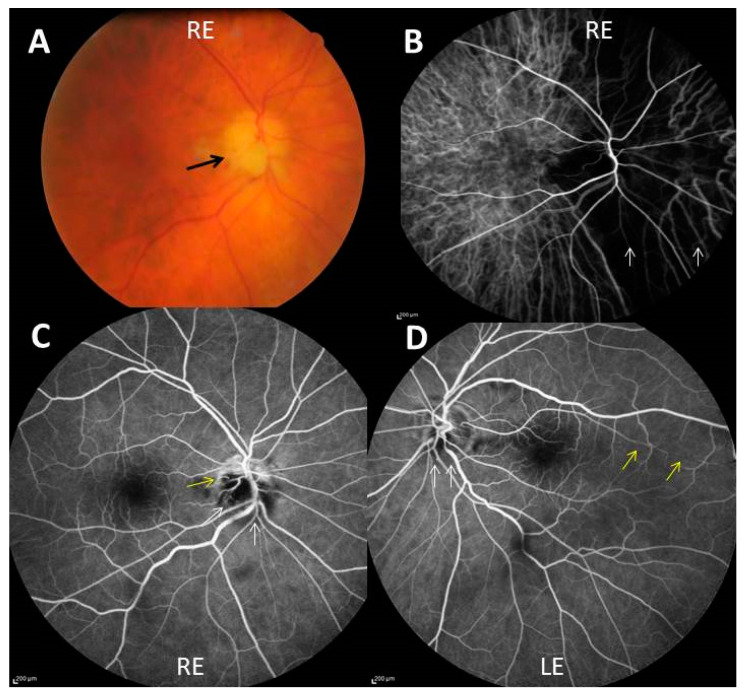
(**A**) Chalky white optic disc edema (black arrow) of the right eye (RE); (**B**) indocyanin green angiography showing severe choroidal hypoperfusion of a large nasal part of the right eye (white arrows); (**C**,**D**) fluorescein angiography showing non perfusion of two-thirds of the optic disc (white arrows), outside an upper crescent, and cilio-retinal artery occlusion (yellow arrow) in the right eye (**C**) and blurring of the lower optic disk margin (white arrows) and hypoperfusion of a long cilioretinal artery (yellow arrows) in the left eye (LE) (**D**).

**Figure 2 jcm-11-01997-f002:**
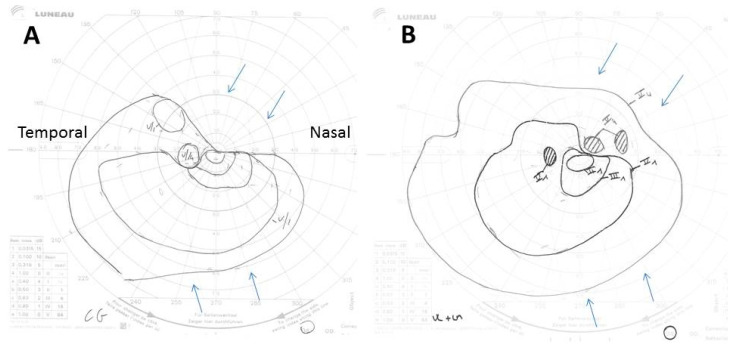
(**A**,**B**) Goldman perimetry showing constriction of the superonasal left visual field (upper arrows) and to a lesser extent of its inferonasal part (lower arrows), 2 days after treatment start (**A**), and its improvement 10 days after its start (**B**). The Roman and Arabic numerals in the images correspond to an increase in size, from 0 to V, and a decrease in intensity, from 4 to 1, of the light signal sent to the patient during the examination to determine the limits of his visual field. Thus, V4 corresponds to the largest and most intense light spot possible.

**Table 1 jcm-11-01997-t001:** Anatomical classification of the ophthalmic lesions of giant cell arteritis GCA.

Anatomical Site	Main Vascular Lesions	Visual Symptoms	Clinical Examination	* Frequency
Orbit				
Eye muscles	Branches of the ophtalmic artery	^†^ Diplopia (all types), ophtalmoplegia	Paresis off one or more extraocular muscles	5–10%
Orbital tissues	Branches of the ophtalmic artery	Diplopia, pain, conjunctival edema	Eye motility disorders, proptosis, chemosis	Rare (≤1%)
Ocular apparatus (from front to back)				
Anterior segment (often generalized ocular ischemia)	Anterior ciliary arteries (ophtalmic artery)	^‡^ Visual loss	Hypotonia, pseudo-uveitis, pupillary abnormalities	Rare (≤1%)
Retina	Central retinal artery	^‡^ Visual loss	Pale edematous retina, cherry red macula, ±cotton wool spots	10%
	^ϕ^ Cilioretinal A. occlusion			22%
Choroid	Posterior ciliary arteries	^‡^ Visual loss	Areas of choroidal infarction	10%
Optic nerve:				
Anterior	Posterior ciliary arteries	^‡^ Visual loss	Shalky white optic disc edema ± peripapillary hemorrhages. Optic atrophy after 4–6 weeks	80–90%
Posterior	Nutrient arteries of optic nerve trunk	^‡^ Visual loss	Normal fundus. Optic atrophy after 4–6 weeks	5%
Occipital brain	Vertebral arteries	^‡^ Visual loss	Homonymous lateral hemianopsia (occipital stroke)	Rare (≤1%)

* Literature-based estimates of the frequency of the different ocular manifestations among GCA patients with ocular involvement [18,19,20,21,22,23]. Some patients may have more than one manifestation. ^†^ Diplopia is most often transient, reported by the patients, with a normal examination by the doctor. ^‡^ Visual loss may be transient (amaurosis fugax) but is most often permanent, of varying degree. ^ϕ^ The only clear estimate of cilioretinal artery occlusion comes from one article; this sign was present in 12 of 55 patients with satisfactory fluorescein angiography [18].

**Table 2 jcm-11-01997-t002:** Rate of permanent visual loss associated with GCA before and during the corticosteroid therapy era.

First Author, Year of Publication [Ref] *	Study Population	Patients with any Degree of Permanent Vision Loss in at Least One Eye		Patients with Complete Blindness in at Least One Eye		Total Number of Completely Blind Eyes
		Total	Bilateral	Total	Bilateral	
	N	N (%)	N (%)	N (%)	N (%)	N (%)
Bruce, 1949 [9]	84	34 (40)	NA	22 (26)	13 (15)	35 (21)
Birkhead, 1957 [14]	^†^ 55	21 (38)	7 (13)	13 (24)	5 (9)	18 (16)
	^‡^ 53	23 (43)	16 (30)	15 (28)	9 (17)	24 (23)
	^ϕ^ 250	121 (48)	92 (37)	67 (27)	54 (22)	121 (24)
Parsons-Smith, 1959 [32]	50	23 (46)	NA	NA	NA	33 (33)
Aiello, 1993 [19]	245	34 (14)	8 (3)	12 (5)	1 (0.4)	14 (6)
Gonzalez-Gay, 2000 [21]	161	24 (15)	8 (5)	NA	NA	NA
Liozon, 2001 [29]	174	23 (13)	NA	NA	3 (14)	NA
Nesher, 2004 [30]	166	32 (19)	NA	NA	NA	NA
Salvarini, 2016 [22]	136	26 (19)	7 (5)	NA	NA	NA
Chen, 2016 [35]	245	20 (8)	6 (2)	4 (2)	2 (1)	6 (2)
Saleh, 2016 [36]	840	85 (10)	13 (2)	18 (2)	2 (2)	20 (2)

* Studies published in 1949, 1957, and 1959 include patients seen before the era of corticosteroid therapy. In Birkhead paper: ^†^ All these patients received corticosteroid therapy after their ocular event, ^‡^ None of these patients received corticosteroid therapy, ^ϕ^ From literature review made by the authors: few patients received corticosteroid therapy. NA: not available.

## Data Availability

Not applicable.

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
