# Peer review of "Ocular Complications of Giant Cell Arteritis: An Acute Therapeutic Emergency"

_jcm, 2022, doi:10.3390/jcm11071997_

Round 1

Reviewer 1 Report

This review introduces ocular complications in GCA, from the histology, causes and development of ocular manifestations, to diagnosis and treatment of GCA with ocular manifestation. The review is in general well-organized and structured and the writing is fine. The topic has great clinical significance, and the content is in line with the topic, and the order makes sense.

Some suggestions:

The authors could dive deeper into the discussion on why GCA patients with ocular manifestations have lower inflammatory parameters (CRP etc): Cid et at 1998, Hernandex Rodriguez 2002 & 2003, van Sleen 2019. Some papers suggested that IL-6, strongly associated with the acute-phase response, might be protective against vision loss, due to its angiogenic capabilities, whereas others present evidence against that. Is it maybe just a shorter symptom duration?

Are the improving percentages of vision loss among patients presenting with GCA associated with the emergence of fast-track clinics?

The inclusion of the case report can add as an interesting example. Please make sure to contextualize the case in the wider field and why you chose to present this case. In addition, this case appears to have a much higher inflammatory response than typical GCA patients with ocular manifestations.

Is there anything known about the effect of tocilizumab on ocular manifestations? Would the fast steroid tapering in patients using tocilizumab not leave them at risk for new ocular relapse?

“as confirmatory examinations either histological or by imaging tests, will not lose sensitivity during the first days of treatment.”
If I’m correct, you did not discuss this in the main body. Please provide references for this statement.

Section 3.1 is very much dependent on ref 15. Do you have any additional references to add to this?

Please make sure to get the alignment of the Table 1 correct, as it is now very hard to read (maybe more an issue for the editor?)

Reviewer 2 Report

General comment :this is a nice paper on ophthalmological features of giant cell arteritis, emphasizing the need for urgent action in not only ocular symptoms but also, more generally, cephalic manifestations. It is interesting that the authors made the effort to look at the history of the disease, but mainly at the pathological findings. It is worth to recall that the vasculitis as such remain extracerebral and is definitely a disorder of the peripheral vessels, before the dura. The paper looks essentially written for ophthalmologists but should be accessible too to internal medicine physicians. The paper is well written and requires minimal English editing.

Specific comments:

  • The authors should not oversee the need for systemic examination, using MRI or 18F-FDG PET/CT for assessing the whole patient. Even if there is an ophtalmological emergency, making a whole-body assessment at the time of diagnosis is essential, for the follow-up evaluation. In this regards, a short paragraph should be added at the end of the manuscript, to highlight the benefit of MRI for morphological evaluation of the disease and 18F-FDG PET/CT for functional evaluation beyond the local cranial situation (as stated in the last EULAR recommendations). In this regards, the delay between the start of corticosteroid therapy and the performance of PET/CT should be limited to a maximum of three days (see Nielsen et. )
  • Figure 1 is beautiful, but Figure 2 is poor and can surely be improved, indicating the nasal and temporal area, and being bigger to clearly show the improvement, in the nasal field.
  • On Figure 1, there is emphasis on the cilioretinal artery, whereas the most ischemic area is just below and may not be related to this particular artery. Difficult to assess without the original fluoangiography. Just check this, please.
  • The use of tocilizumab should be more extended, as it is ‘step 2’ of the therapeutic approach according to the EULAR 2018 recommendations.

Minor comments

  • Table 1 in its current format is hard to read: lines should be aligned.
